# Contamination, Source Identification, Ecological and Human Health Risks Assessment of Potentially Toxic-Elements in Soils of Typical Rare-Earth Mining Areas

**DOI:** 10.3390/ijerph192215105

**Published:** 2022-11-16

**Authors:** Jiajia Fan, Li Deng, Weili Wang, Xiu Yi, Zhiping Yang

**Affiliations:** 1Key Laboratory of Subsurface Hydrology and Ecological Effects in Arid Region, Ministry of Education, Chang’an University, Xi’an 710064, China; 2Ecological Environment Planning and Environmental Protection Technology Center of Qinghai Province, Xining 810007, China; 3Third Institute of Oceanography, Ministry of Natural Resources, Xiamen 361005, China; 4Jiangxi Research Academy of Ecological Civilization, Nanchang 330036, China

**Keywords:** potentially toxic elements, mining areas, source allocation, human health risks

## Abstract

The mining and leaching processes of rare-earth mines can include the entry of potentially toxic elements (PTEs) into the environment, causing ecological risks and endangering human health. However, the identification of ecological risks and sources of PTEs in rare-earth mining areas is less comprehensive. Hence, we determine the PTE (Co, Cr, Cu, Mn, Ni, Pb, Zn, V) content in soils around rare-earth mining areas in the south and analyze the ecological health risks, distribution characteristics, and sources of PTEs in the study area using various indices and models. The results showed that the average concentrations of Co, Mn, Ni, Pb and Zn were higher than the soil background values, with a maximum of 1.62 times. The spatial distribution of PTEs was not homogeneous and the hot spots were mostly located near roads and mining areas. The ecological risk index and the non-carcinogenic index showed that the contribution was mainly from Co, Pb, and Cr, which accounted for more than 90%. Correlation analysis and PMF models indicated that eight PTEs were positively correlated, and rare-earth mining operations (concentration of 22.85%) may have caused Pb and Cu enrichment in soils in the area, while other anthropogenic sources of pollution were industrial emissions and agricultural pollution. The results of the study can provide a scientific basis for environmental-pollution assessment and prevention in rare-earth mining cities.

## 1. Introduction

The rapid development of industrialization and the process of agricultural intensification have inevitably led to the influx of potentially toxic elements (PTEs) into the environment [1], while the contamination of soil by PTEs has been the focus of scholars’ attention due to its complexity, insidiousness, difficulty in degradation, and toxicity [2]. Over time, PTEs gradually become accumulated in the soil, further leading to the erosion of soil nutrients and the degradation of biological functions [3], and they have negative effects on farmland yield and ecological health [4]. Meanwhile, certain highly migratory or toxic elements can accumulate and be transferred to edible parts of plants or enter the food chain, thereby endangering human health through dietary intake [5,6]. Previous studies have shown that many biological diseases are associated with overexposure to PTEs. For instance, Pb exposure can lead to the disruption of the respiratory and neurological systems [7] and inflammatory cells and mediators associated with a variety of diseases [8]. Although Cu is an essential element, excessive intake can cause gastrointestinal discomfort (nausea, vomiting, abdominal pain) and anemia [9], in addition to inducing changes in cell activity [10]. Hence, the evaluation of the contamination levels and ecological health risks associated with PTEs in the study area was imperative.

Currently, the contents of PTEs in soil are mainly influenced by natural factors as well as being adjusted by anthropogenic inputs [11,12]. It was assumed that PTEs in soil were primarily derived from natural sources [13], such as weathering of parent soil [14], and anthropogenic sources, which included mining activities, transportation and agricultural use of fertilizers and pesticides [15,16,17]. In particular, the extraction and smelting of raw and auxiliary rare-earth materials can also enrich PTEs in the soil, caused by the presence of PTEs [18,19]. It was noted that Cu, Zn, Pb, and Cr were significantly higher than their background levels around some rare-earth mining areas [20]. Specifically, in southern ionic mines, potentially toxic-element ions were leached from disordered piles of mine spoils in response to rainfall [21]. A study has shown that the water quality in the area where the mine is located is better during the dry season than the rainy season [22]. The wastewater and waste from beneficiation procedures, such as in-situ leaching technologies, and cellar fumes migrate, adsorb and coalesce with the atmosphere and water flow, which also aggravates the enrichment of PTEs on the surface of cities [23,24]. The migration of PTEs from different human activities has a bidirectional pattern of vertical penetration and lateral flow, and the migration capacity of elements in each layer is variable. All characteristics determined that the distribution of PTEs will show diversity. So, it should be useful to explore the variation of spatial distribution and sources of PTEs in typical mining areas.

Positive matrix factorization (PMF) [25,26,27] was adopted in the early 1990s as a commonly available receptor model for the source allocation of atmospheric particulate matter [28], and in subsequent years, it was also utilized effectively to apportion sources from sediment, soil, and aquatic systems [29,30,31]. The PMF model does not confine the source contribution to non-negative values and does not necessitate the identification of specific sources, just an uncertainty input at each sampling point [32]. Consequently, PMF was employed in combination with correlation analysis to allocate the sources of PTEs, based on the distribution of PTEs, and the main natural and anthropogenic sources in the area were determined.

The study area is located in a mining city in Jiangxi Province, which is rich in rare-earth resources and has acidic soils with high ionic activity. In recent years, the region has additionally promoted the forestry industry and agricultural intensification, possibly influencing soil levels of PTEs. More attention has been paid in recent years to studies on vegetation damage and erosion from mining processes, and landscape breakage. However, studies on PTEs entering the environment due to in-situ leaching and causing human health risks also deserve attention. Therefore, the main objectives of this research were to: (1) assess the levels and distribution of potentially toxic-element pollution in topsoil in the study area; (2) quantify the sources of potentially toxic elements in soil as assigned using a PMF model and correlation analysis; (3) evaluate the ecological and human health risks posed by potentially toxic elements in soil. This study mainly discusses the distribution pattern and pollution sources of PTEs in typical rare-earth mining cities, which can provide support for the management of mining areas and the restoration of the surrounding ecological environment.

## 2. Materials and Methods

### 2.1. Study Area

The sampling site is situated in the county of southern Gannan, Jiangxi Province, upstream of the Ganjiang River in the Yangtze River system, covering 2350 km^2^. The main soil type is red soil. The sampling region is a humid monsoon climate zone on the southern edge of the central subtropics, with a mild climate and plentiful rainfall, an annual average temperature of 20.1 °C and annual precipitation of 1548.3 mm. Based on data obtained from the World Uniform Soil Database [33], the soils sampled in the study area were classified as alisols, anthrosols and acrisols. The region has a flourishing mining industry, focused on rare-earth metals, tungsten, coal and uranium, which represents its principal economic activity. Agricultural cultivation and production are steadily increasing, and primary crops include rice, soybeans, navel oranges and watermelon.

### 2.2. Site Description and Sample Collection

According to the principle of uniform distribution of points, sampling was carried out at 31 points across the study area in 2021 (Figure 1). For each sampling site, soil was collected with a wooden shovel from the surface layer (0–20 cm) of the five plume-distributed sub-sampling sites. The soil from the five sites was thoroughly mixed and the composite sample was placed in a labeled sealed bag. The exact coordinates of each site were recorded by a highly sensitive handheld GPS. The soil samples were placed in a cool room without direct sunlight. After natural air drying, the soil samples were processed promptly.

Soil samples were ground through a 2 mm nylon sieve after the removal of animal and plant residues and stones, then digested with HNO_3_ (65–68%)-HCl (36–38%)-HF (40%)-HClO_4_ (70–72%) in the microwave ablation instrument (Multiwave PRO, Anton Paar, Austria) to allow estimation of the pseudo-total contents of PTEs. After the microwave digestion was completed, the samples were put into the acid pick-up apparatus at 180°. When the samples became clear and free of impurities, they were transferred into a stoppered cuvette and the volume was fixed to 25 mL. The contents of Co, Cr, Cu, Mn, Ni, Pb, Zn and V were measured by inductively coupled plasma-optical emission spectrometry (ICP-OES, America) with detection limits of 0.013, 0.013, 0.013, 0.054, 0.054, 0.013, 0.04 and 0.013 μg/L, respectively. GBW07447(GSS18) was used as a reference sample for quality control, and blank samples were obtained to minimize the intrusion of background obstacles.

In addition, a portion of the soil was passed through 1 mm and 0.075 mm nylon sieves for the determination of soil pH (pH meter, PHS-3E, REX, Shanghai, China), soil particle size (laser particle size analyzers, Mastersizer 2000, Malvern, Malvern city, UK), and total soil carbon and total nitrogen (C/N elemental analyzer, vario MACRO cube, Elementar, Hanao, Germany).

### 2.3. Pollution Index Evaluation

The potential ecological risk index (RI) was established to evaluate soil ecological risk [34]. This method allows the comprehensive evaluation of the risks posed by PTEs in soil based on the nature and environmental behavior of these PTEs, not only taking into account PTE levels but also incorporating the ecological, environmental and toxic effects of PTEs [35,36]. The calculation formula is as follows:(1)$$\mathrm{RI}=\sum E_{r}^{i}\;;\;E_{r}^{i}=T_{r}^{i}\times \frac{C_{i}}{S_{i}}$$

In which, RI is the potential ecological risk index of various PTEs in soil, $E_{r}^{i}$ is the potential ecological risk coefficient of the *i*-th PTE; $T_{r}^{i}$ is the toxicity response coefficient of the *i*-th PTE. *C_i_* is the concentration of *i*-th PTE. The measured content of *S_i_* is the background value of the *i*-th PTE, and the background value of Jiangxi soil was used in this study [37].

The overall potentially toxic-element contamination status of the soil was assessed using the Nemerow integrated pollution index (NPI) [38,39]. The NPI formula is as listed below:(2)$$\mathrm{NPI}=\sqrt{\frac{PI_{i-max}^{2}+PI_{i-mean}^{2}}{2}}$$

Here, *PI_i-max_* and *PI_i-mean_* are the maximum and average values of potentially toxic-element contamination index *PI*, respectively. The *PI* calculation formula is as follows, where *PI*, *C_i_* and *S_i_* represent the contamination index, content and background values of metals, respectively.
(3)$$PI=\frac{C_{i}}{S_{i}}$$

### 2.4. Positive Matrix Factorization and Correlation Analysis

In this study, positive matrix factorization (PMF) was applied in combination with correlation analysis for the source assignment of PTEs in soil [25]. Correlation analysis is an effective method for exploring relationships between multiple variables; the stronger the correlation between two elements, the stronger the homology between these two elements [40,41]. In this study, the Pearson correlation coefficient [42] was used to calculate correlations between different elements in the study area.

Positive matrix factorization was developed based on the theory of factor analysis, which treats data about the potentially toxic-element levels in soil as a matrix and then decomposes this matrix into a factor contribution matrix and a factor component matrix based on least-squares limited iterations, to quantitatively resolve potential pollution sources and their respective contribution rates [43,44]. PMF does not necessitate knowledge of the emission inventory of pollution sources or environmental parameters regarding pollutant dispersion and transfer, effectively avoiding deviations caused by topographical and meteorological factors [45,46]. The USEPA PMF 5.0 program was used to analyze eight potentially toxic-element sources and allocate the contributions of each source; the choice of factor number was key to the model.

### 2.5. Human Health Risk Assessment

Health risk assessment relates the level of contaminants in the environment to the potential for toxic effects on humans. This study evaluated the non-carcinogenic risks associated with seven PTEs (all except Mn) in adults and children. The health risk assessment approach used was defined by the U.S. Environmental Protection Agency (EPA) for calculating exposure dose and performing risk assessments via exposure [47]. The three primary routes of human exposure to contaminants in soil include direct ingestion, dermal contact, and inhalation of vapors. As presented in Appendix A, the average daily dose (ADD) of PTEs through each pathway can be calculated using a series of equations [48]. The non-carcinogenic risk of each PTE can be expressed as the total hazard quotient (HQi), which is the sum of the hazard quotients (HQADD) associated with each of the three exposure routes [49]. HI is calculated as the sum of the HQi values, representing the overall non-carcinogenic risk of the site, and the evaluation standards of HI was shown in Appendix A.

### 2.6. Statistical Analysis

In this study, a geostatistical approach was applied to characterize the spatial distribution of PTEs [50]. The chosen method was the inverse distance interpolation method, and the software was ArcGIS 10.8.1 (ESRI, 2021). The statistical packages Origin 2021 (OriginLab, Northampton, MA, USA) and Office Excel 2016 (Microsoft Corporation, Redmond, WA, USA) were used for statistical calculations and correlation analysis of all data.

## 3. Results and Discussion

### 3.1. Contents of Potentially Toxic Elements and Property of Soil

A summary of the statistical characteristics of potentially toxic-element levels in soil samples, as well as background values for Jiangxi soil and Chinese soil pollution standards, are shown in Table 1. Except for Pb and Zn, which slightly exceeded primary standards, all PTEs had average concentrations that did not exceed the primary or secondary standards for agricultural soils in China, indicating that the site does not pose a hazard to human health at this time. The average contents of Co, Mn, Ni, Pb, and Zn were significantly more than the background values of Jiangxi soil, exceeding these values by 1.49, 1.28, 1.16, 1.41 and 1.62 times, respectively. However, Cr, Cu and V did not exceed background levels. Across all soil samples, background values were exceeded by 56.25% for Co, 15.63% for Cr, 43.75% for Cu, 56.25% for Mn, 56.25% for Ni, 62.50% for Pb, 90.63% for Zn, and 9.38% for V. According to Table 1, all potentially toxic elements except Zn had high coefficients of variation (CV > 40%) and wide concentration ranges (Figure 2), indicating that most of the soil inputs for potentially toxic elements in this study area were associated with anthropogenic sources.

Further, the study calculated the enrichment factor (EF) of PTEs in soil samples to respond to the extent of anthropogenic disturbance in the study area [53,54]. The average EF values of PTEs in soil samples were as follows: 1.09–8.57 for Co, 0.58–3.67 for Cr, 0.37–9.47 for Cu, 0.35–11.20 for Mn, 0.60–6.23 for Ni, 0.43–12.21 for Pb, 0.923–16.49 for Zn and 0.51–1.39 for V (Appendix A). In the study area, all elements except Cr, V and Cu showed enrichment. In particular, Pb and Zn exhibited significant pollution levels in some samples. The specific grade evaluation methods are shown in Appendix A.

In addition, the pH range of the soils collected in the study area was 4.4.–7.7, with an overall weak acidity. The ranges of total nitrogen and total carbon in the soils of the study area were 0.2–1.7 g/kg and 1.1–21.8 g/kg. Soil texture is a natural property of soil stability that can reflect the parent material source and certain characteristics of the soil formation process. In the collected samples, the clay grain content ranged from 1.4 to 19.7%. According to the national classification standards, most of the soils in the study area are sandy soils, and a few of them are loamy types. Correlation analysis of PTE concentrations and physicochemical properties of soils in the study area showed weak and insignificant correlations (Appendix A). As shown by the pH distribution map (Appendix A), the overall trend was high in the east and low in the west, and lower values of pH appear in the areas with higher population density and more frequent anthropogenic activities. Although pH was mainly influenced by natural factors such as soil parent material and geological background, anthropogenic factors cannot be ignored. Moreover, the high values of TC and TN were mainly distributed in woodlands and grasslands, and lower levels were found in places with higher anthropogenic utilization.

### 3.2. Spatial Distribution of Potentially Toxic Elements

The spatial variation in potentially toxic-element content in soil samples across the study area is shown in Figure 3. There was a high degree of similarity between the spatial distributions of Co and Pb in soil; both were concentrated in the northwest portion of the study area, although the concentrations of Co and Pb diverged in some areas. Hot spots for these two PTEs were mainly distributed along both sides of the highway, and this area was surrounded by mining sites and was densely populated [55]. Traffic pollution has been reported as a major source of Pb [56,57]. Mining inevitably increases levels of PTEs in the surrounding area.

As illustrated in Figure 3, Cr, Cu, and Ni were predominantly spread in the central and western parts of the study area. Potentially toxic elements were enriched in the center of the study area, which were densely populated and enriched for anthropogenic activities and agricultural production. Some soil in the enrichment area was cropland with frequent agricultural activity, and agricultural products such as commercial fertilizers, animal manure, pesticides and fungicides contain large amounts of Cu [6,58,59]. Therefore, agricultural pollution may be one of its sources. The southwest region of the enrichment site contained several rare-earth ore sites as well as porcelain ore, sulfur iron ore and copper ore, which may be a major contributor to PTE pollution. Previous studies have shown that one of the sources of Cr contamination in soil is the soil’s parent material [60,61].

Sites enriched for Mn and V in the study area were dispersed in a punctate pattern with minimal variation. The lowest exceedance rate (9.38%) was reported for V, and its average content did not exceed background values, indicating that levels of V in this area may be controlled primarily by the soil parent material [62]. However, we did identify two hot spots for V in the west and north, potentially due to multiple nearby mining sites. Hot spots for Mn were found in the southernmost and central parts of the study area, and a survey of the area revealed several local manufacturing plants, suggesting that Mn levels may be influenced by industrial pollution [63].

The distribution of Zn was different from those of other potentially toxic elements. Sites with high concentrations were primarily positioned in the southern part of the study area, and the sampling sites exceeded the standard by 90.63%, indicating that Zn may be influenced by different sources. In addition to the distribution of industrial enterprises across the study area, a Jiulong wind farm was located in the southern hot spot. Exhaust gas generated by its working process may settle into the soil and affect Zn levels [64]. Secondly, the transportation process also generates Zn-containing particles [65].

### 3.3. Pollution Index Analysis

The Nemerow integrated pollution index and potential ecological risk index were utilized to assess contamination levels of PTEs in soil in the study area. The average single-factor pollution indexes of PTE in soil were ranked as follows: Zn (1.62) > Co (1.49) > Pb (1.41) > Mn (1.28) > Ni (1.16) > Cu (0.94) > Cr (0.67) > V (0.57) (Table 2). Among these, Cr and V were at safe levels, Cu was at a safe level but trending towards light pollution, and Zn, Co, Pb, Mn, and Ni were at levels associated with light pollution. However, among the overall data, some sampling points in the study area exhibited heavy pollution levels (*P_i_* > 3). In particular, Co and Pb had maximum values of 7.67 and 6.74, respectively, indicating anthropogenic disturbance in some portions of the site.

Concrete results of the NPI are shown in Table 2, and the spatial interpolation method was applied to draw an evaluation map of the NPI for PTEs in soil (Figure 4a). The figure showed that NPI was high in the western part of the study area, and we identified two hot spots where the index indicated a heavy pollution level (NPI > 3). Specific evaluation criteria are shown in Appendix A. Hot spot locations were distributed on both sides of the highway, so traffic pollution may represent a major risk of pollution to the area. In combination with the single-factor index, the highest contributors were identified as Pb and Zn. In addition, there were many mining areas nearby, and the disorderly stacking of solid waste, dust discharge and acid mine wastewater from mine development can cause metal ions to diffuse into the soil [66,67,68].

The $E_{r}^{i}$ values of eight PTEs in soil in the study area can be ranked as follows: Co > Pb > Cu > Cr > Ni> Zn > Mn > V (Table 2). On the basis of combining the original grading standards [34], this study was improved with reference to the research of Fernández et al. [69] and Li et al. [70]. Specific grading standards are shown in Appendix A. In the study area, Co, Cr and Pb had high potential risk factors in some regions and were at moderate pollution levels ($E_{r}^{i}$ > 20), while Cu, Mn, Ni, Zn and V had low ecological risks in all areas. The RI distribution patterns for potentially toxic elements in the study area, as shown in Figure 4b, exhibited a similar distribution trend to NPI, with a gradual decrease in overall risk from west to east. We identified a moderate risk area (RI > 50) in the western region of the study area, indicating that this site may be influenced by anthropogenic impacts. We found a moderate risk area in the eastern part of the study area. This area had the highest population density, and the majority of land was arable. It is speculated that agricultural pollution may be the main cause (Appendix A).

### 3.4. Quantitative Distribution of Sources of Potentially Toxic Elements in Soil

The sampling site concentration data and uncertainty data files were utilized as input data for PMF. Distinct positions of the rotation parameter Fpeak were studied, as well as the choice of three, four, or five factors for the operation. Q**_Robust_** was close to Q**_True_** when four factors were used, and the best fit was achieved between the true content value and the model prediction. The determination coefficients showed a strong correlation between all potentially toxic elements (R^2^ for Ni was 0.86, and the R^2^ values of all elements were greater than 0.500), and all PTEs showed strong run classification. These results indicate that the PMF software has good overall resolution, and the number of factors selected can best explain the information contained in the measured data.

The results obtained from EPA PMF software analysis are shown in Figure 5. The figure shows that the main loadings of Factor 1 were Pb, Zn and Cu, with respective contributions of 76.66%, 39.3% and 31.7% (Appendix A). According to a correlation analysis of PTEs (Figure 6), the correlation between Pb and Cu was 0.32, indicating that they may share a common source [71]. Previous studies in surrounding areas have shown that traffic pollution was one of the main causes of Pb pollution in soil [72]. Although the use of leaded gasoline was banned worldwide, potentially toxic elements still exist in the soil because they do not migrate easily, and Pb from vehicle exhaust was discharged into the atmosphere through fine particles [73,74,75]. Tire friction and braking while driving can also cause the release of Pb, Zn and Cu, which are reported to contribute to road dust at levels up to 10.7%, 38.1% and 55.3%, respectively [76,77]. According to the PTEs distribution map (Figure 3), a highway was built in an area enriched for Pb, Zn and Cu, especially Pb (Appendix A). There were rich rare-earth resources in the study area, and the tailings and residues generated during the exploitation process were deposited on the surface in an uncontrolled manner [78]. During the rainy season in the southern region, PTEs in the waste were diffused to the surrounding area through rainwater drenching and acidic wastewater, enriching the associated elements into the soil [79,80]. Around a rare-earth mining area in Guangdong Province, the soil Pb content can reach 79.33–134.54 mg/kg [81], and some studies have shown that the correlation between rare-earth elements and Pb and Cu was strong [73,82,83]. Therefore, Factor 1 was defined as a possible source of rare-earth mining and traffic pollution.

Factor 2 was mainly dominated by Mn, with a contribution of 70.6%, while the contributions of Co and Zn reached 28.2% and 33.1%, respectively (Figure 5). Although Mn is often associated with natural sources due to less interference from human activities [84], the coefficient of variation of Mn at sampling sites in the study area was large, and its maximum concentration was 1402 mg/kg, far exceeding the background value of soil in Jiangxi Province. Therefore, it was speculated that levels may be influenced by human activities. Analysis of the study area showed several processing plants, including smelters and mop processing plants, in the area enriched for Mn, Zn, and Co. Therefore, industrial production may be the source, in agreement with a study by Jiang et al. [85]. Therefore, Factor 2 may correspond to industrial sources.

Factor 3 was mainly dominated by Cu, Co, and V, with the rate of contribution of Cu reaching 52.8% (Figure 5). The correlation analysis clearly showed a significant positive correlation between Co, Cu, and V, indicating that these three elements may have the same source. Previous evidence has shown that the use of pesticides and fertilizers in agricultural farming has a significant effect on Cu levels in soil and represents an important factor in soil accumulation of Cu [59,86,87]. Field investigation revealed that forestry is vigorously developed in this area, fruit trees such as navel orange trees, peach trees and plum trees are planted, and some land in the copper hotspot area represents cropland. Therefore, Factor 3 probably represents a mixed source of agricultural pollution and mining extraction emissions.

Factor 4 accounted for 26.54% of the total variance, and Cr and Ni exhibited heavy loading of 52.7% and 52.4%, respectively (Figure 5). Compared with other elements in the study area, Cr and Ni showed lower CV (45.51%, 44.49%) and lower spatial variability. Moreover, only 15.63% of soil samples exceeded background levels of Cr, and its average concentration did not exceed the background value. These findings indicated that Cr and Ni were less subject to anthropogenic disturbance. Correlation analysis showed that Cr and Ni were significantly correlated (*p* < 0.001), with a correlation coefficient of 0.73 (Figure 6). Weathering of rocks was the primary natural source of Cr in soil, and naturally occurring concentrations of Cr in soil range from 10 to 50 mg/kg depending on the mineralogy of the parent material. Indeed, in ultramafic soils, Cr levels can vary from 634 to 125,000 mg/kg [88,89,90]. Previous case studies had also shown that natural sources were an important contributor to Cr and Ni levels in soils, with concentrations more similar to natural background levels [17,91]. Therefore, Factor 4 may represent natural origins of the soil parent material.

### 3.5. Health Risk Assessment of Potentially Toxic Elements in Soils

The current study examined the non-carcinogenic risks of PTEs to humans through three exposure pathways, and the results are shown in Appendix A. The rank order of different exposure pathways for PTEs in soil in this study area was (for both adults and children) direct intake > dermal contact > oral and nasal inhalation, consistent with previous studies [92,93]. Overall, the highest intake was Cr and Pb. Children were more likely to experience unintentional ingestion or exposure to PTEs, such as through pica behavior or finger-sucking [94], and children may be more heavily exposed to PTEs during outdoor activities [95].

Table 3 lists the non-carcinogenic-risk PTEs in the study area for adults and children. Based on non-carcinogenic risk levels of single PTEs to adults, the order of HQ for the different elements was V > Pb > Cr > Cu > Ni > Co > Zn (Table 3). However, none of these values exceeded 1, indicating that the PTEs in this area did not cause a non-carcinogenic risk to adults. For the overall regional non-carcinogenic risk index (HI) (adults and children), the main non-carcinogenic factors among PTEs in the study area were Cr, Pb and V. The sum of HQ of the three elements accounted for more than 90% of HI (Figure 7). The HI of children was higher than that of adults, but the HI indexes of all sampling sites were below 1. Individual site indexes reached 0.8 or more, indicating that no non-carcinogenic health risk was currently posed to people by potentially toxic-element contamination of soil at this site.

## 4. Conclusions

This study aimed to evaluate the risks and sources of potentially toxic elements in a typical rare-earth mining area in southern China. In this study area, most PTEs exceed the soil background values with high coefficients of variation. Spatially, the PTEs were distributed with high heterogeneity, which may be influenced by the surrounding highways and mining activities, and the potential ecological risk index and Nemerow pollution index were consistent with their distribution patterns. Among them, the single risk values of Nemerow index and potential ecological index were larger for Co, Mn, Cu, Pb and Zn, with the largest values of 7.67 and 38.34. Positive matrix factorization and correlation analysis revealed four main sources of potentially toxic elements, namely, traffic pollution and rare-earth mining, industrial pollution, agricultural pollution, and weathering of soil parent material. The Pb from traffic pollution and rare-earth mining contributed the most to the non-carcinogenic risk at the site, accounting for 26.81% and 40.35% of the risk for adults and children, respectively, and more attention should be paid to pollution prevention in the future. Notably, the non-carcinogenic risks for all elements showed higher levels in children than in adults, indicating that children were more vulnerable to PTEs. In conclusion, this research provided an effective method for quantifying apportioned risk, which is essential for pollution control and risk reduction in resource-limited study areas.

## Figures and Tables

**Figure 1 ijerph-19-15105-f001:**
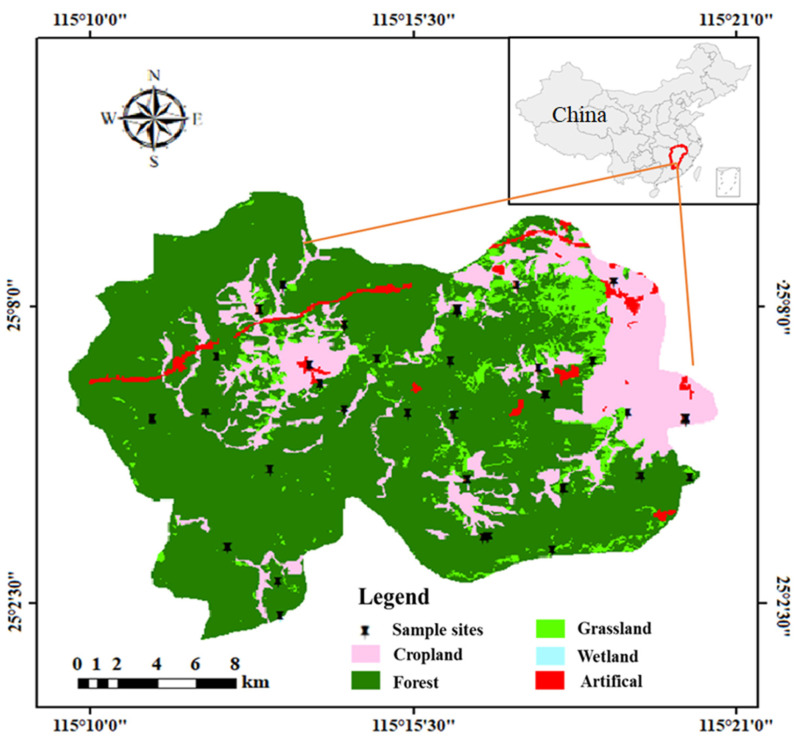
Location of the study area and sampling sites.

**Figure 2 ijerph-19-15105-f002:**
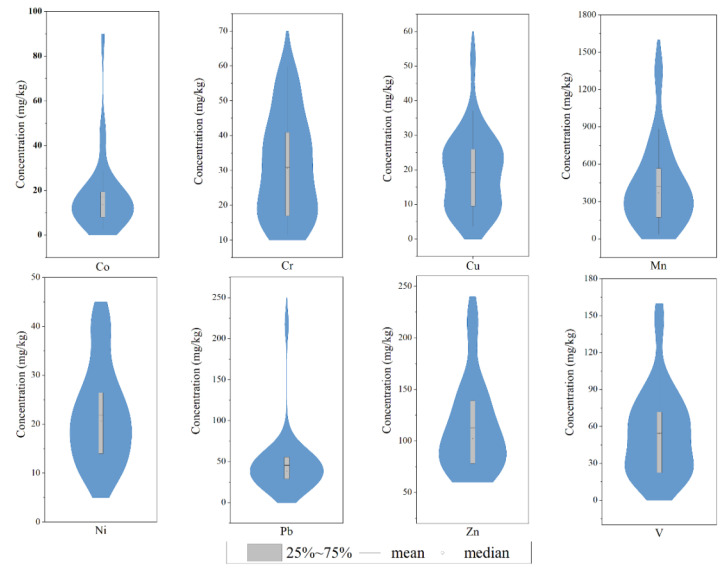
Pseudo-total contents of potentially toxic elements in soil in the study area. Solid lines indicate the mean values determined in this study; circles indicate the median values determined in this study.

**Figure 3 ijerph-19-15105-f003:**
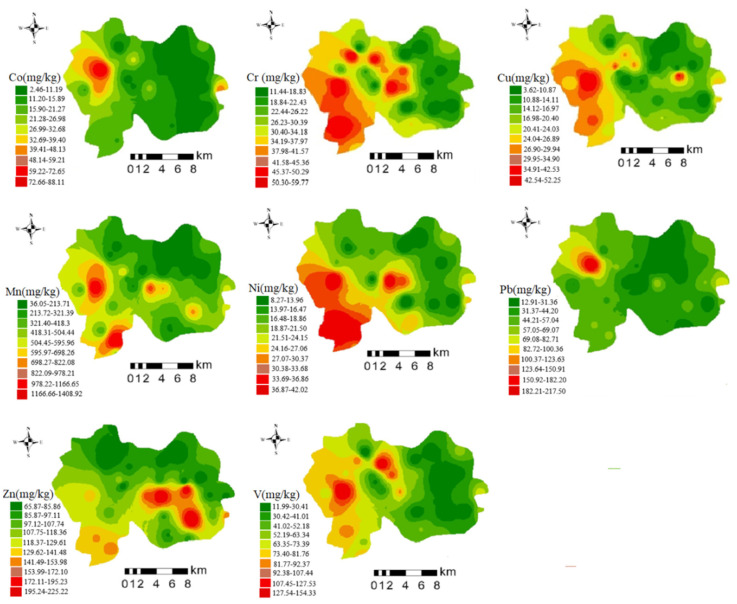
Spatial distribution of potentially toxic-element concentrations in soil.

**Figure 4 ijerph-19-15105-f004:**
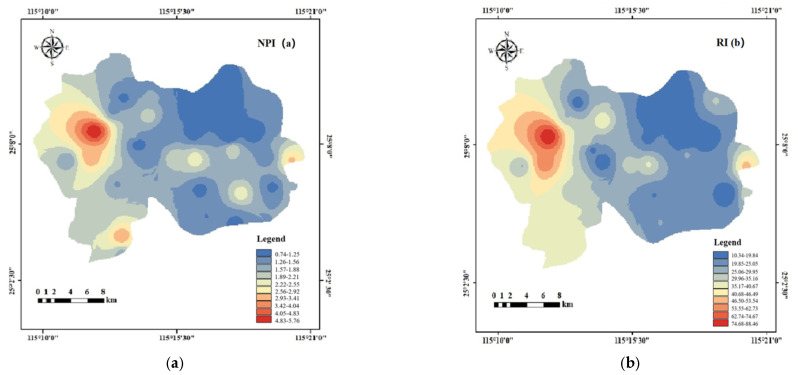
Class distribution of soil potential ecological risks in the study area. (**a**) Distribution of Nemerow integrated pollution index for soils in the study area. (**b**) Distribution of potential ecological risks index for soils in the study area.

**Figure 5 ijerph-19-15105-f005:**
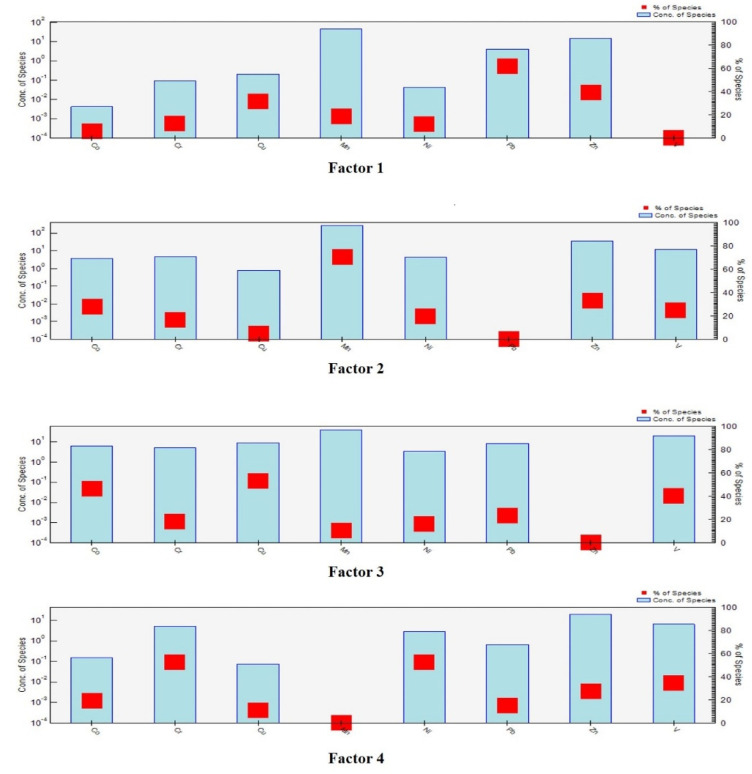
Factor profiles and origin contributions of potentially toxic elements in soil based on PMF.

**Figure 6 ijerph-19-15105-f006:**
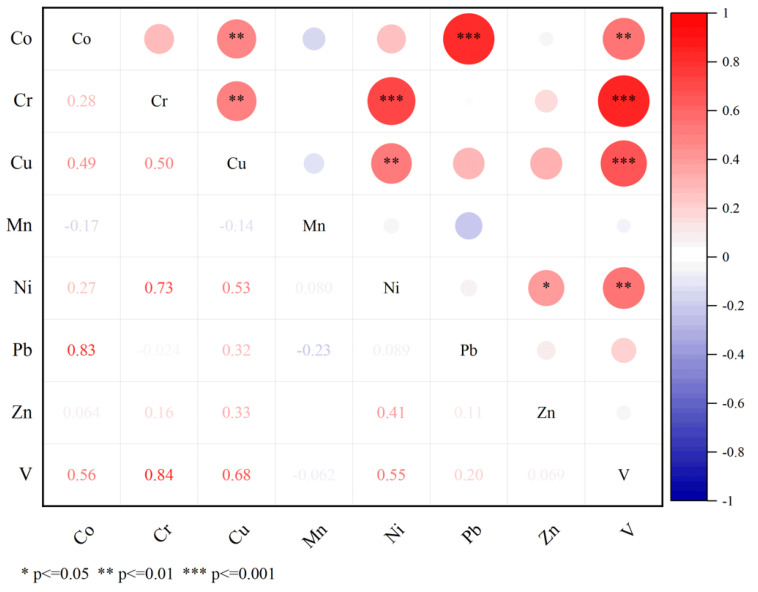
Correlation analysis of potentially toxic elements in soil in the study area. * represents the probability significance level 0.05, ** probability significance level 0.01, and *** represents the probability significance level 0.001.

**Figure 7 ijerph-19-15105-f007:**
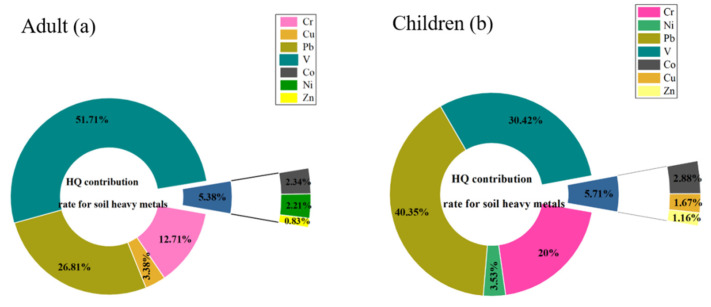
Contribution of different potentially toxic elements to non-carcinogenic risk.

**Table 1 ijerph-19-15105-t001:** Descriptive statistics of potentially toxic-element concentrations (mg/kg).

Elements (mg/kg)	Co	Cr	Cu	Mn	Ni	Pb	Zn	V
Mean	17.21	30.91	19.17	418.88	21.86	45.41	112.65	54.39
Maximum	88.19	59.82	52.29	1409.22	42.04	217.61	225.25	154.46
Minimum	2.46	11.42	3.61	35.42	8.26	12.87	65.83	11.93
CV(%)	93.61	45.51	56.13	77.85	44.49	77.54	37.81	63.43
Background ^a^	11.5	45.9	20.3	328	18.9	32.3	69.4	95.8
Average of China ^b^	13	61	23	582	26.9	27	74.2	82.4
Grade I ^c^	-	90	35	-	40	35	100	-
Grade II ^c^	-	150	50	-	40	250	200	-

^a^ Background: the background soil values of Jiangxi Province [36]. ^b^ Data taken from Teng, 2014 [51]. ^c^ Grade: Soil environmental quality standard (GB15618-1995) [52].

**Table 2 ijerph-19-15105-t002:** Nemerow integrated pollution index and potential ecological risk index of potentially toxic elements in soil in the study area.

	*Pi*				$E_{r}^{i}\;$			
	Maximum	Minimum	Mean	CV(%)	Maximum	Minimum	Mean	CV(%)
Co	7.67	0.21	1.49	93.62	38.34	1.07	7.48	93.62
Cr	1.30	0.25	0.67	45.52	26.07	4.98	13.5	45.52
Cu	2.58	0.18	0.94	56.13	12.9	0.89	4.72	56.13
Mn	4.30	0.11	1.28	77.86	4.30	0.11	1.28	77.86
Ni	2.22	0.44	1.16	44.50	4.45	0.87	2.31	44.50
Pb	6.74	0.39	1.41	77.55	33.69	1.99	7.03	77.55
Zn	3.25	0.95	1.62	37.81	3.25	0.95	1.62	37.81
V	1.61	0.12	0.57	63.43	3.22	0.25	1.14	63.43
NPI	5.76	0.73	1.75	56.46	-	-	-	-
RI	-	-	-	-	106.64	7.37	28.95	53.89

**Table 3 ijerph-19-15105-t003:** Non-carcinogenic risk index of potentially toxic elements in adults and children.

Elements	Adult				Child			
	HQ_ing_	HQ_der_	HQ_inh_	HQ	HQ_ing_	HQ_der_	HQ_inh_	HQ
Co	1.30 × 10^−3^	6.03 × 10^−6^	4.80 × 10^−4^	1.70 × 10^−3^	8.25 × 10^−3^	2.94 × 10^−5^	7.98 × 10^−4^	9.08 × 10^−3^
Cr	9.20 × 10^−3^	2.20 × 10^−4^	1.70 × 10^−4^	9.60 × 10^−3^	5.93 × 10^−2^	3.38 × 10^−3^	2.82 × 10^−4^	6.29 × 10^−2^
Cu	8.10 × 10^−4^	1.80 × 10^−3^	7.61 × 10^−8^	2.60 × 10^−3^	4.97 × 10^−3^	2.76 × 10^−4^	1.27 × 10^−7^	5.24 × 10^−3^
Ni	1.60 × 10^−3^	3.86 × 10^−5^	3.17 × 10^−6^	1.70 × 10^−3^	1.05 × 10^−2^	5.97 × 10^−4^	2.75 × 10^−7^	1.11 × 10^−2^
Pb	1.90 × 10^−2^	9.10 × 10^−4^	2.06 × 10^−6^	2.00 × 10^−2^	1.24 × 10^−1^	2.34 × 10^−3^	3.43 × 10^−6^	1.30 × 10^−2^
Zn	5.60 × 10^−4^	6.63 × 10^−5^	5.96 × 10^−8^	6.00 × 10^−4^	3.60 × 10^−3^	5.13 × 10^−5^	9.93 × 10^−8^	3.65 × 10^−3^
V	1.20 × 10^−2^	2.75 × 10^−2^	2.56 × 10^−6^	3.90 × 10^−2^	7.50 × 10^−3^	2.10 × 10^−3^	2.05 × 10^−6^	9.57 × 10^−2^
HI		8 × 10^−2^			3.1 × 10^−1^			

## Data Availability

Not applicable.

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
