# Peer review of "Contamination, Source Identification, Ecological and Human Health Risks Assessment of Potentially Toxic-Elements in Soils of Typical Rare-Earth Mining Areas"

_ijerph, 2022, doi:10.3390/ijerph192215105_

Round 1
Reviewer 1 Report
I think the topic of this manuscript is interesting and important, and the urban environment state assessment, because of the threat to human health, should be a priority in the development of modern interdisciplinary bioindication methods. The use of bioindicators is a simple and complementary approach to physico-chemical contaminants extraction procedures. To get a better understanding of relationships between potentially toxic elements concentrations in the environment and their health impact, it is, therefore, reasonable to assess health risk for both children and adults in accordance with the Environmental Protection Agency procedure.
I have no objections to the methodology or description of the research area. Everything is written clearly and is not in doubt. Also, the results are presented in an understandable way. The paper has an appropriate structure, is well-written, and contains up-to-date references. Conclusions are consistent with the evidence, and they address the main questions posed.
Therefore, I have no major reservations and comments, apart from those mentioned in the attachment.

Reviewer 2 Report
Many of the justifications for the high metal content in soils are based on the proximity of mines. It would be important for maps to show the location of mines.
The sampling process is essential to guarantee the quality of the results. In this work the sampling process is not described. It should be described.
The classification of pollution indexes is based on a particular scale for each of the indexes and described in the original literature. These scales are not shown. Should be presented when the indexes, worked with, and their calculation formulas are presented.
Conclusions can be improved.
References are numbered at the end but in the text they are referred to by the name/date system. They must be harmonized.
Line 13: toxic elements (PTEs) … All the elements considered PTE and studied in this work must be listed.
Line 15: The phrase needs revision, too long: ”Hence, in this study, we took the soils around typical ionic rare earth mining areas in the south, determined the contents of PTEs in soils and analyzed the spatial distribution pattern by geostatistical approach, and performed ecological health risk evaluation and Positive matrix factorization (PMF) traceability analysis to assess the impact of rare earth mining on the surrounding environment.”
Sugestions:
we took –> we collect
around typical ionic rare earth mining areas -> around rare earth mining areas
determined the contents … -> we determine the content…
Line 21: not homogeneous, and the hot -> not homogeneous and the hot
Line 62: Rephrase, double negative: where the mines were located did not meet water quality standards in the wet season, but not in the dry season
Line 109: Fig. 1: it would be interesting to mark active mines on the map.
Line 140: reference is necessary : and the background value of Jiangxi soil was used
Line 193, Table 1: The data “Average of China” must have a reference or an evaluation method description.
Line 272: Fig 3: it would be interesting to mark active mines on the maps.
Line 384, Fig. 5: Remove minor ticks from % scale.
Line 399: Table 3: The line below adults and children must have a gap between both classes. The data must be written in textual scientific numbering and not in computer scientific numbering.
Author Response
Please see the attachmentnt

Reviewer 3 Report
The research work is interesting. However, the authors must present some thematic maps proposed in the comments and perform map algebra to have greater certainty in their results and conclusions.

Reviewer 4 Report
The paper „ Ecological Risk Distribution and Sources Identification of 2 Potentially Toxic Elements Pollution in Typical Rare Earth Mining Areas” is interesting, concerns the aspect of heavy metal contamination of soils. The content of elements in soil is due to mineral composition (parent materials) or is the result of their anthropogenic contamination. However, after examining the content, I have a few comments, the fulfilment of which will determine whether the work will be released for printing. The authors do research on soils, but the reader doesn’t know anything about them?
Lack of systematic analysis of soils. I propose to publish the classification of the soils studied according to WRB. However in the article the information on basic physical and chemical properties of analyzed soil samples is missing altogether. In my opinion the knowledge of soil textural properties, pH, organic carbon and total nitrogen content is inevitable for interpreting chemical composition of soils, PTE in particular. In my opinion, it is a mistake to adopt a single geochemical background value for such a large area. Therefore, unless this requirement is fulfilled I cannot recommend this manuscript for publication in the journal. The authors draw far-reaching conclusions based on the total content of potentially toxic elements in the analysed soils.
The total content of potentially toxic elements in soil depends on many factors and does not indicate the current level of environmental hazard. This risk is mainly related to their bioavailability, since they are associated with the solid phase of the soil.
Round 2
Reviewer 3 Report
The information presented by the authors in alternate files responds to my observations made in the previous version.
Reviewer 4 Report
Recommendation to the Authors:
Unfortunately, the authors did not meet my expectations: - there’s no soil taxonomy (WRB or FAO) of the analyzed area.
Also the information on the range of values concerning the basic physical and chemical properties of the analyzed soils, don’t carry enough to the presented work. Please add for example basic statistics on the spatial differentiation of these properties.
